# Extraction, Characterization, and Nutraceutical Potential of *Prosthechea karwinskii* Orchid for Insulin Resistance and Oxidative Stress in Wistar Rats

**DOI:** 10.3390/foods13152432

**Published:** 2024-08-01

**Authors:** Gabriela Soledad Barragán-Zarate, Luicita Lagunez-Rivera, Alfonso Alexander-Aguilera, Rodolfo Solano, Gerard Vilarem

**Affiliations:** 1Laboratorio de Extracción y Análisis de Productos Naturales Vegetales, Centro Interdisciplinario de Investigación para el Desarrollo Integral Regional Unidad Oaxaca, Instituto Politécnico Nacional, Hornos 1003, Santa Cruz Xoxocotlán C.P. 71230, Oaxaca, Mexico; gbarraganz@ipn.mx (G.S.B.-Z.); asolanog@ipn.mx (R.S.); 2Facultad de Bioanálisis, Universidad Veracruzana, Carmen Serdán s/n, Col. Flores Magón, Veracruz C.P. 91700, Veracruz, Mexico; 3Escuela de Medicina, Universidad Cristóbal Colón, Carretera Veracruz-Medellin s/n, Col. Puente Moreno, Boca del Río C.P. 94271, Veracruz, Mexico; 4Laboratoire de Chimie Agro-Industrielle, ENSIACET, 4 Allée Emile Monso, BP 44362, 31030 Toulouse, Cedex 4, France; gerard.vilarem@ensiacet.fr

**Keywords:** reactive oxygen species, HOMA-IR, TyG index, SOD, CAT

## Abstract

*Prosthechea karwinskii* is an endemic orchid of Mexico with cultural significance for its ornamental, food, religious, and medicinal uses. In traditional medicine, diabetic patients use the leaves of this plant to lower glucose levels. The present study evaluated the effect of *P. karwinskii* leaves extract on the antioxidant enzymes superoxide dismutase (SOD) and catalase (CAT) in a model of obese rats with insulin resistance for its nutraceutical potential to reduce insulin resistance and oxidative stress. Obesity and insulin resistance were induced with 40% sucrose in water for 20 weeks. Four groups (control rats, obese rats, obese rats administered the extract, and obese rats administered metformin) were evaluated. Extract compounds were identified by UHPLC-ESI-qTOF-MS/MS. Glucose, insulin, triglyceride, and insulin resistance indices (HOMA-IR and TyG), as well as the activity of the antioxidant enzymes, increased in rats in the obese group. Administration of *P. karwinskii* extract and metformin reduced glucose, insulin, triglyceride, and insulin resistance indices and antioxidant enzyme activity to values similar to those of the control group. Therefore, this study shows the nutraceutical potential of *P. karwinskii* extract as an ingredient in the formulation of dietary supplements or functional foods to help treat diseases whose pathophysiology is related to oxidative stress and insulin resistance.

## 1. Introduction

Insulin resistance (IR) is a metabolic disorder associated with obesity, where the sensitivity and cellular response to insulin in tissues is decreased, as well as glucose metabolism. IR is involved in the pathophysiology of metabolic disorders such as diabetes, dyslipidemia, non-alcoholic fatty liver disease, and cancer. Insulin resistance is characterized by elevated levels of reactive oxygen species (ROS). The origin of ROS being mainly mitochondrial, hyperglycemia can cause alterations to mitochondrial morphology and stimulate different mitochondrial enzymatic pathways, such as nicotinamide adenine dinucleotide phosphate oxidase (NADPH oxidase), nitric oxide (NO) synthase uncoupling, xanthine oxidase, and lipoxygenases, cyclooxygenases, and peroxidases. In turn, increased levels of oxidative stress with increased ROS production cause insulin resistance. Oxidative stress and insulin resistance are increased in obese individuals [1].

Oxidative stress participates in the pathogenesis of many chronic diseases, including IR and diabetes, and occurs due to an imbalance between elevated ROS generation and/or poor antioxidant defense. Due to their antioxidant activity, medicinal plants are an alternative for the treatment of various diseases [2].

*Prosthechea karwinskii* (Mart.) J.M.H. Shaw is an endemic orchid of Mexico that is used in the Mixtec region of the state of Oaxaca to decorate church altars during Semana Santa celebrations as well as to prepare various local dishes and in traditional medicine [3]. The extract of its leaves decreased glucose levels [4] and IR [5] in rats with metabolic syndrome and inhibited ROS in an *ex vivo* model with peripheral blood mononuclear cells [6]. Its effect on markers of inflammation and the activity of the enzyme superoxide dismutase (SOD) was evaluated in a model of acute gastric injury, where no clear involvement of SOD in the gastric protection conferred by the extract was observed [7]. However, nothing has been published on the effect of the orchid extract on other parameters related to oxidative stress in an *in vivo* model that would give us more information on the impact of the extract when administered for longer periods. Considering this background, this study aimed to evaluate the effect of *P. karwinskii* leaves extract on the antioxidant enzymes SOD and catalase (CAT) in a model of obese rats with insulin resistance to determine its nutraceutical potential with respect to insulin resistance and oxidative stress and thereby assess its potential as an ingredient in the formulation of food supplements and functional foods to help treat diseases related to insulin resistance and oxidative stress.

## 2. Materials and Methods

### 2.1. Materials, Reagents, and Kits

The following commercial kits were obtained: a Catalase assay Kit (707002, Cayman chemical, Ann Arbor, MI, USA), a Superoxide Dismutase assay Kit (706002, Cayman chemical, Ann Arbor, MI, USA), and an Insulin assay Kit (589501, Cayman Chemical, Ann Arbor, MI, USA). Phosphate-buffered saline (PBS), distilled water, formic acid, acetonitrile, and other reagents were purchased from Sigma Aldrich (Toluca, Mexico), and analytical kits for glucose and triglycerides were purchased from Spinreact (Naucalpan, Mexico).

### 2.2. Plant Material

The leaves of *P. karwinskii* used in this study were collected in 2017 in Zaachila, Oaxaca (16°57′ N latitude, 96°45′ W longitude; 1490 m altitude), dehydrated whole, and stored in paper bags at room temperature away from light and humidity until extraction. Species determination was performed by one of the authors (RS), and a backup specimen was deposited in the OAX herbarium (Solano 4037). In this study, *Prosthechea* is recognized as the accepted generic name of the orchid genus, in agreement with Pridgeon et al. [8], Soto et al. [9], Villaseñor [10], and Solano et al. [11].

### 2.3. Obtaining the Extract

The extract of the *P. karwinskii* leaves was obtained by an ultrasound-assisted method, following Barragan-Zarate et al. [6] (2020), in a 750 W ultrasound processor (VCX 750, Scientific SENSE). The solvent used was 50% ethanol in water with a sample–solvent ratio of 1:18 (g:mL), and the extraction temperature was maintained at 40 °C with a water bath for 20 min. The extract was stored in an amber glass bottle that was sealed and kept at freezing temperature until analysis to avoid degradation of its compounds.

### 2.4. Compound Identification with UHPLC-ESI-qTOF-MS/MS

For the identification of plant compounds, the method described by Barragan-Zarate et al. [5] was followed. An ultra-high-performance liquid chromatography (UHPLC) system (Ultimate 3000, Thermo Fisher Scientific, Waltham, MA, USA) combined with an Impact II mass spectrometer (Bruker) with electrospray ionization (ESI) and quadrupole time of flight (qTOF) was used. The column used was the Thermo Scientific Acclaim 120 C18 (2.2 μm, 120 Å, 50 × 2.1 mm), and an elution gradient was used with the following mobile phases: A: 0.1% formic acid in water and B: acetonitrile, with a flow rate of 0.35 mL/min and a temperature of 25 °C. The gradient system was as follows: 0% B (0–2 min), 1% B (2–3 min), 3% B (3–4 min), 32% B (4–5 min), 36% B (5–6 min), 40% B (6–8 min), 45% B (8–9 min), 80% B (9–11 min), 0% B (12–14). The mass spectrometer was operated in negative electrospray mode at 0.4 Bar (5.8 psi), in autoMSMS, with a mass range of 50–700 *m*/*z*. The data obtained were processed with Bruker’s DataAnalysis software 3.1. Compounds were identified by comparing their exact masses and fragmentation patterns with those collected from libraries and scientific articles.

### 2.5. Induction of Obesity

A total of 24 Wistar rats, 21 days old, were divided into 2 groups: a control group (CG, *n* = 6) and an obese group (OG, *n* = 18). The rats were fed *ad libitum* with standard food, the CG group drank drinkable water and the OG group drank water with 40% sucrose to induce obesity, and the induction period was 20 weeks, as reported by Barragan-Zarate et al. [5]. The rats were maintained under normal conditions in the biotherium of the Escuela de Medicina of the Universidad Cristóbal Colón, following the official Mexican standard for the care and handling of animals, NOM-062-ZOO-1999 [12], as well as the international norms and standards for the care and use of laboratory animals. The experimental protocol was approved by the Cuerpo académico Genómica y Salud UVE CA-317 of Facultad de Bioanalisis from the Universidad Veracruzana with registration GS-317-1-2010 on August 2022.

### 2.6. Experimental Design

After 20 weeks of obesity induction, the obese group (OG) was divided into 3 groups: the OG group, which received no treatment; the PK group, which was administered *P. karwinskii* extract at a dose of 300 mg/kg; and the MET group, which was administered metformin at a dose of 200 mg/kg. The selection of the doses was based on a previous study [5], where the effective dose for controlling the evaluated metabolic syndrome parameters was 300 mg/kg. Treatments were administered with an oral feeding cannula. Rats in the OG, PK, and MET groups continued to drink 40% sucrose water during the experimental diet. Comparisons were made with the CG group (the group in which obesity was not induced and which did not receive any treatment). The experimental period was 4 weeks.

### 2.7. Evaluated Parameters

At the end of the experimental period, the rats were fasted for 12 h and sacrificed. The anesthetic used was sodium pentobarbital. The weight of the rats was recorded, as well as the weight of the total adipose tissue. Blood was collected to determine glucose, insulin, and triglyceride levels and to calculate indicators of insulin resistance. The Homeostatic Model Assessment of Insulin Resistance (HOMA-IR) was calculated as described by Matthews et al. [13] with Equation (1):HOMA-IR = [fasting insulin (μIU/mL) × fasting glucose (mmol/L)]/22.5. (1)

The triglyceride–glucose index (TyG) was calculated according to Simental-Mendía and Guerrero-Romero [14] with Equation (2):TyG = Ln [fasting triglycerides (mg/dL) × fasting glucose (mg/dL)]/2. (2)

In addition, liver samples were taken and homogenized for the determination of the antioxidant enzymes SOD and CAT, following the respective kit manufacturers’ instructions.

### 2.8. Statistical Analysis

Results are reported as means ± standard deviations (*n* = 6). Analysis of variance (ANOVA) was performed, followed by Neuman’s Keul test. Differences were considered statistically significant at *p* < 0.05.

## 3. Results and Discussion

### 3.1. Compounds Identified with UHPLC-ESI-qTOF-MS/MS

Table 1 shows the information on the compounds identified in the extract of *P. karwinskii* with UHPLC-ESI-qTOF-MS/MS, which were quinic acid, malic acid, succinic acid, L-(-)-phenylalanine, guanosine, neochlorogenic acid, chlorogenic acid, rutin, kaempferol-3-O-ruthinoside, azelaic acid, pinellic acid, and embelin.

The tentative identification of the compounds was performed by comparing their spectral data (*m*/*z* values of the precursor ions and fragmentation patterns) with data reported in the literature and in databases (the MassBank and Bruker’s MetaboBase libraries). The values reported by De Souza et al. [15] were used for the tentative identification of quinic acid, with a precursor ion *m*/*z* value of 191.0575 and fragment 127 corresponding to the neutral loss of formic acid (−46 Da); chlorogenic acid, with a precursor ion *m*/*z* value of 353.0890 and fragments 179 and 191 corresponding to caffeic acid, and quinic acid due to the loss of caffeoyl (−162 Da); and rutin, with a precursor ion *m*/*z* value of 609.1490 and fragment 301 corresponding to the rutinoside associated with the internal fragmentation of the flavonoid glycoside. Data reported by Ma et al. [16] were employed for the identification of quinic acid, with a precursor ion *m*/*z* value of 191.0553 and fragments of 111.0072 and 127.0389; neochlorogenic acid, with a precursor ion *m*/*z* value of 353.0886 and fragments of 179.0349 and 191.0562; and rutin, with a precursor ion *m*/*z* value of 609.1502 and fragments of 300.0266 and 301.0341. Data reported by Dahibhate et al. [17] were employed for the identification of azelaic acid, with a precursor ion *m*/*z* value of 187.0974 and fragments of 169, 125.1, and 97; pinellic acid, with a precursor ion *m*/*z* value of 329.2333 and fragments of 229.1 and 171.1; and embelin, with a precursor ion *m*/*z* value of 293.1758. Compounds identified via libraries were compared with spectral data from standards analyzed with similar instruments with the same type of ionization and fragmented with similar collision energies.

### 3.2. Effect of P. karwinskii Extract on Obesity, Insulin Resistance, and Oxidative Stress Parameters

Table 2 shows the weight record of the rats at weeks 0 and 20 of induction (before the experimental diet) and at the end of the experiment, along with indices of insulin resistance at week 20 as well as the total adipose tissue values of the rats at the end of the experimental diet. This information proves that the rats in the OG group were obese and presented insulin resistance before starting the experimental diet. 

Figure 1 shows the effect of *P. karwinskii* extract on glucose, insulin, and triglyceride levels, as well as on HOMA-IR and TyG insulin resistance indices. All these parameters presented higher values in the OG group compared to the CG group. In the case of Figure 1a,c,e, only the OG group was different from the CG group. In the case of Figure 1b,d, the OG, PK, and MET groups were different from the CG group; however, PK differed from OG in both cases, while MET differed only in the case of Figure 1d. Administration of the extract and metformin reduced glucose and triglyceride levels and insulin resistance indices compared to the OG group. As for insulin, only the extract had an effect compared to OG. Table 3 shows the weekly liquid and solid diet and caloric intakes of the groups during the experimental diet period. Higher fluid intake was observed in the OG, PK, and MET groups compared to the CG group. Regarding the solid diet, consumption was higher in the CG group compared to the OG, PK, and MET groups. Although there were no significant differences between the total caloric intakes of the OG, PK, and MET groups at the end of week 4, there were significant differences in glucose and triglyceride levels and insulin resistance indices, as well as body weight and adipose tissue, evidencing the beneficial effect of the extract despite consumption of the same sucrose-rich diet.

Other studies have reported similar results, such as that of Paunovic et al. [18], where rats were fed a high-fat, high-fructose diet supplemented with blackcurrant juice and there was no significant difference in the caloric intake between the different groups; however, the supplemented diet did reduce adipose tissue and triglyceride levels and improved glucose tolerance, though, unlike our extract, it did not affect glucose or insulin levels. The study by Mostafa et al. [19] evaluated *Vitis vinifera* seed extract in rats fed a diet rich in fat and carbohydrates, and the extract reduced glucose and triglyceride levels and weight gain compared to the untreated group.

*P. karwinskii* leaves extract was able to decrease glucose levels and insulin resistance. The compounds identified in the extract that could have been responsible for the effect of the extract on parameters related to glucose metabolism were chlorogenic acid [20] and embelin [21], which regulate glucose metabolism and decrease IR [20], and rutin, which decreases carbohydrate absorption, inhibits gluconeogenesis, increases tissue glucose uptake, increases insulin secretion by β-pancreatic cells, and protects the islets of Langerhans [22]. Given the relationship of IR with various metabolic disorders and the ability of *P. karwinskii* extract to attenuate them, it has the potential to help treat diseases in which IR is implicated.

Figure 2 shows the activity of the antioxidant enzymes SOD and CAT, which were increased in the OG group compared to the GC group. However, in the PK and MET groups, the activity of these enzymes was downregulated to values that did not differ significantly from those recorded for the GC group.

Hyperglycemia can cause altered mitochondrial morphology and increase cellular oxidation of glucose and production of nicotinamide adenine dinucleotide (NADH), which contributes to oxidation in cellular processes, thereby increasing ROS generation [1]. ROS are produced in cells during normal aerobic metabolism; the generated ROS are removed by antioxidant enzymes, including SOD and CAT, which protect cells against oxidative stress by cellular detoxification of O_2−_ and H_2_O_2_, as excessive production of ROS and/or deficient antioxidant capacity can generate oxidative stress [2]. Hyperglycemia also activates different metabolic signaling pathways, triggering the activation of the antioxidant system to reduce ROS production and oxidative stress [1].

In the present study, the activity of the evaluated antioxidant enzymes increased in the obese group under oxidative stress compared to the control group. Although this behavior differs from the common behavior of antioxidant enzyme activity under oxidative stress, similar behavior has been reported in other investigations, such as Chen et al.’s [23] review on oxidative stress in non-alcoholic fatty liver disease and Barbosa et al.’s review [24] on oxidative stress in hypertensive diseases of pregnancy, where, in some cases, an increase in the activity of the antioxidant enzymes CAT, SOD, and glutathione peroxidase was observed, suggesting a compensatory mechanism for the increase in ROS due to oxidative stress.

Similarly, the research of Zeng et al. [25] on the effect of cadmium stress on the antioxidant system of crabs showed that SOD activity increased in both the hepatopancreas and the intestine in the cadmium-stressed group compared to the control group, while in the case of CAT, its activity decreased in the hepatopancreas but increased in the intestine in the Cd-stressed group. So, they consider antioxidant enzymes as non-specific adaptive mechanisms that protect against oxidative damage and serve as the first line of defense against oxidative stress.

According to Aouacheri et al. [26], whose results showed that SOD activity in diabetic patients was increased compared to the control group, increased antioxidant enzyme activity stimulates the cellular ability to remove and limit damage caused by ROS, acting as an adaptive response and a compensatory mechanism to detoxify harmful metabolites related to oxidative stress [23,26]. Leghi et al. [27] also observed that SOD enzyme activity was increased in a group with non-alcoholic steatohepatitis compared to those without, suggesting that the increased SOD activity counteracted the overproduction of O_2-_ radicals.

The increase in CAT activity could occur in response to high SOD activity by the synergistic action of both enzymes [2], as it increases H_2_O_2_ production by SOD, which is subsequently catalyzed by CAT. The extract downregulated the activity of SOD and CAT enzymes, which increased in OG to values like those presented by CG, possibly through a decrease in ROS, the levels of which were increased by hyperglycemia and IR. In addition, a previous study showed the ability of *P. karwinskii* leaf extract to inhibit ROS in cells [6]. This suggests that the compounds acted to reduce ROS, maintaining redox balance, without the physiological need to increase antioxidant enzyme activity. In other words, they can neutralize ROS and produce more stable radicals [28].

There have been other investigations where the effects of evaluated compounds were found to be similar to those reported in this research, such as that of Barbosa et al. [28], where they evaluated the effect of acai pulp in pregnant rats fed a high-fat diet and the activities of the antioxidant enzymes SOD and CAT were increased in a group under oxidative stress compared to the control group and, similar to what happened with our extract, the activity of antioxidant enzymes was modulated and biomarkers of oxidative stress which had been altered by the high-fat diet were reduced, thus improving the oxidative balance.

Something similar also occurred in okra plants: when they were stressed with heavy metals such as cadmium, ROS production increased [29]. It was observed that the activity of the antioxidant enzymes SOD and CAT increased in Cd-stressed plants compared to non-stressed plants and that receiving treatments with chelating agents, such as malic acid and EDTA, increased the resistance of plants to oxidative stress by decreasing H_2_O_2_ and the activity of SOD and CAT enzymes. Possibly something like what happens in plants occurred in our model, since, according to Hussain et al. [30], they develop defensive mechanisms to minimize or protect themselves from the toxic effects of ROS, maintaining a balance between their synthesis and decomposition to preserve cellular redox hemostasis. Antioxidant enzyme activity is a key indicator of the antioxidant defense mechanism.

The compounds identified in the extract with antioxidant properties are rutin [31], kaempferol-3-O-ruthinoside [32], embelin [21,33], and guanosine [34]. According to reports, these compounds decrease oxidative stress.

## 4. Conclusions

The activity of the antioxidant enzymes SOD and CAT was increased in the obese group as an adaptive response or compensatory mechanism to minimize the overproduction of ROS due to hyperglycemia and IR, which generated oxidative stress. Administration of the extract downregulated antioxidant enzyme activity to values like those of the control group, possibly by decreasing ROS formation by regulating glucose and IR levels, without increasing antioxidant enzyme activity to reduce oxidative stress. The orchid *P. karwinskii* has nutraceutical potential to treat conditions whose pathophysiology involves oxidative stress and/or insulin resistance. Thus, it can be considered a potential ingredient in the formulation of dietary supplements and functional foods to help treat diseases related to insulin resistance and oxidative stress. Therefore, future research concerning *P. karwinskii* should be directed towards the study of the properties of the extract as an ingredient, as well as the design of functional foods and nutraceuticals that share its properties.

## Figures and Tables

**Figure 1 foods-13-02432-f001:**
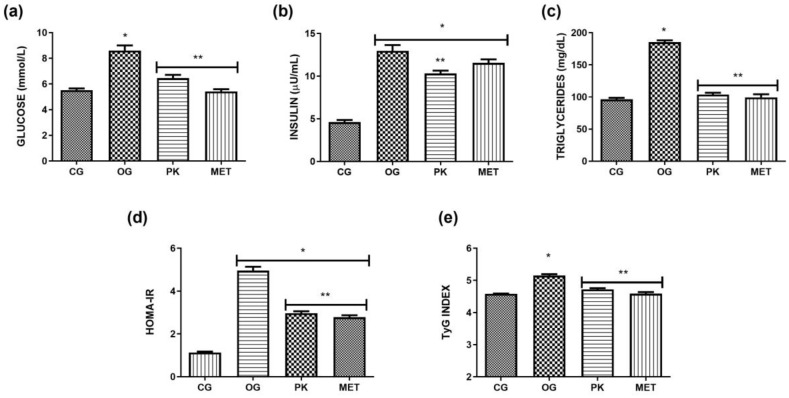
Glucose (**a**), insulin (**b**), and (**c**) triglyceride levels and HOMA-IR (**d**) and TyG (**e**) insulin resistance indices. Values expressed as means ± SDs. * Indicates a significant difference (*p* ˂ 0.05) with respect to CG; ** indicates a significant difference (*p* ˂ 0.05) with respect to OG. CG: control group, OG: obese rats, PK: obese rats that received extract, and MET: obese rats that received metformin.

**Figure 2 foods-13-02432-f002:**
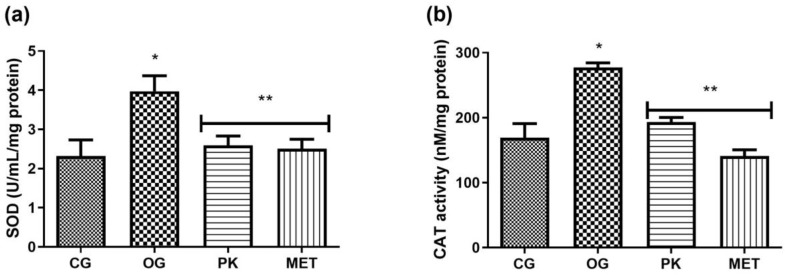
Superoxide dismutase enzyme activity (SOD) (**a**) and catalase enzyme activity (CAT) (**b**). Values expressed as means ± SDs. * Indicates a significant difference (*p* ˂ 0.05) with respect to GC; ** indicates a significant difference (*p* ˂ 0.05) with respect to OG. CG: control group, OG: obese rats, PK: obese rats that received extract, and MET: obese rats that received metformin.

**Table 1 foods-13-02432-t001:** Chemical structures and information on compounds identified in leaves extract of *Prosthechea karwinskii* with UPLC-ESI-qTOF-MS/MS.

Peak	RT (min)	*m*/*z*[M-H]^−^	Error (ppm)	MS/MS Fragments	Compound (Chemical Formula)	Type of Compound	Relative Yield (%)	Chemical Structure
1	0.7	191.0557	1.9	85.0293, 87.0078, 111.0443, 127.6945	Quinic Acid ^abd^(C_7_H_12_O_6_)	Cyclitol, cyclic polyol, and cyclohexanecarboxylic acid	22.21	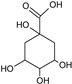
2	0.8	133.0140	1.8	115.0032	Malic acid ^d^(C_4_H_6_O_5_)	Dicarboxylic organic acid	5.71	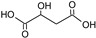
3	1.2	117.0191	2.3	73.0290, 99.0072	Succinic acid ^d^(C_4_H_6_0_4_)	Dicarboxylic organic acid	1.89	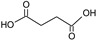
4	2.3	164.0712	3.2	72.0072, 103.0539, 147.0442	L-(-)-Phenylalanine ^e^(C_9_H_11_NO_2_)	Amino acid	0.53	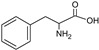
5	2.7	282.0833	3.9	108.5347, 133.0157, 150.0429	Guanosine ^e^(C_10_H_13_N_5_O_5_)	Nucleoside	0.46	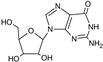
6	6.0	353.0867	3.1	179.0365, 191.0556	Neochlorogenic acid ^b^(C_16_H_18_O_9_)	Caffeoylquinic acid, phenolic compound	4.35	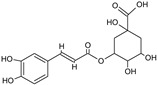
7	6.3	353.0866	3.4	173.0452, 179.0365, 191.0556	Chlorogenic acid ^a^(C_16_H_18_O_9_)	Caffeoylquinic acid, phenolic compound	8.44	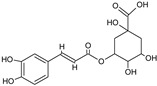
8	6.5	609.1438	3.8	300.0266, 301.0335	Rutin ^abe^(C_27_H_30_O_16_)	Flavonoid glycoside	15.52	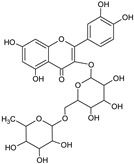
9	6.6	593.1489	3.9	284.0314, 285.0393	Kaempferol-3-O-rutinoside ^e^(C_27_H_30_O_15_)	Flavonol glycoside	25.30	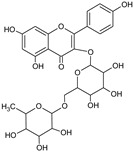
10	6.8	187.0970	3.1	97.0653, 125.0963, 169.0889	Azelaic acid ^c^(C_9_H_16_0_4_)	Dicarboxylic acid	2.10	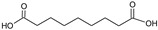
11	8.2	329.2321	3.9	171.1023, 229.1436	Pinellic acid ^c^(C_18_H_34_O_5_)	Carboxylic acid	1.09	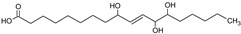
12	11.5	293.2112	3.6	275.2013, 235.1680, 223.1685	Embelin ^c^(C_17_H_26_O_4_)	Para-benzoquinone	8.60	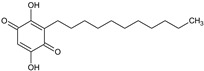

Superscripts correspond to supporting references for compound identification: ^a^ De Souza et al. [15], ^b^ Ma et al. [16], ^c^ Dahibhate et al. [17], ^d^ MassBank library, ^e^ Bruker’s MetaboBase library. RT: retention time.

**Table 2 foods-13-02432-t002:** Body weight record for Wistar rats at weeks 0 and 20 of obesity induction and at the end of the experiment, along with total adipose tissue.

	CG	OG	PK	MET
Weight at week 0 (g)	74.3 ± 17.9	78.2 ± 15.5	-	-
Weight at week 20 (g)	368.5 ± 38.1	456.2 ± 42.6 *	-	-
TyG index at week 20	7.73 ± 0.48	8.85 ± 0.53 *	-	-
HOMA-IR at week 20	1.73 ± 0.25	5.01 ± 0.47 *	-	-
Weight at the end of the experiment (g)	423.1 ± 30.2	527.0 ± 45.3 *	487.0 ± 42.8 #	489.0 ± 55.2 #
Total adipose tissue at the end of the experiment (g)	14.3 ± 3.1	37.92 ± 4.4 *	29.38 ± 3.5 * **	33.35 ± 4.9 *

Values expressed as means ± SDs (*n* = 6). * Indicates a significant difference (*p* ˂ 0.05) with respect to CG; ** indicates a significant difference (*p* ˂ 0.05) with respect to OG; # indicates that the value does not differ significantly with respect to either CG or OG. CG: control group, OG: obese rats, PK: obese rats that received extract, and MET: obese rats that received metformin.

**Table 3 foods-13-02432-t003:** Weekly record of weight, liquid, and food consumption, as well as caloric intake, for each of the experimental diet groups.

Week 1	CG	OG	PK	MET
Weight (g)	389.00 ± 28.70	479.00 ± 35.97 *	472.00 ± 32.49 *	494.00 ± 67.77 *
Liquid consumption (mL/day)	30.60 ± 4.59	17.42 ± 2.61 *	13.85 ± 2.07 *	9.85 ± 1.47 * **
Liquid consumption (mL/day/100 g)	7.86 ± 0.58	3.63 ± 0.27 *	2.93 ± 0.20 * **	1.99 ± 0.27 * **
Equivalent in kcal in drinkable water	0.00	13.58 ± 1.02 *	10.96 ± 0.75 * **	7.44 ± 1.02 * **
Feed consumption (g/day)	24.28 ± 3.64	17.57 ± 2.63 *	17.71 ± 2.65 *	18.42 ± 2.76 *
Feed consumption (g/day/100 g)	6.24 ± 0.46	3.66 ± 0.27 *	3.75 ± 0.25 *	3.73 ± 0.51 *
Equivalent in kcal in feed	19.35 ± 1.42	11.37 ± 0.85 *	11.63 ± 0.80 *	11.56 ± 1.58 *
Total Kcal/day/100 g body weight	19.35 ± 1.42	24.95 ± 1.87 *	22.59 ± 1.55 *	19.01 ± 2.60 **
**Week 2**	**CG**	**OG**	**PK**	**MET**
Weight (g)	398.00 ± 25.21	482.00 ± 33.55 *	459.00 ± 35.41 *	482.00 ± 55.16 *
Liquid consumption (mL/day)	32.57 ± 4.88	28.14 ± 4.22	33.60 ± 5.04	30.77 ± 4.61
Liquid consumption (mL/day/100 g)	8.18 ± 1.10	5.83 ± 0.95 *	7.32 ± 0.38 **	6.38 ± 0.06 *
Equivalent in kcal in drinkable water	0.00	65.39 ± 4.55 *	76.80 ± 5.92 * **	75.08 ± 10.15 * **
Feed consumption (g/day)	27.28 ± 4.09	16.00 ± 2.4 *	11.85 ± 1.77 * **	12.71 ± 1.90 * **
Feed consumption (g/day/100 g)	6.85 ± 0.44	3.31 ± 0.23 *	2.58 ± 0.19 * **	2.63 ± 0.35 * **
Equivalent in kcal in feed	21.25 ± 1.36	10.29 ± 0.71 *	8.00 ± 0.61 * **	8.17 ± 1.10 * **
Total Kcal/day/100 g body weight	21.25 ± 1.36	75.68 ± 5.26 *	84.80 ± 6.54 *	83.26 ± 11.25 *
**Week 3**	**CG**	**OG**	**PK**	**MET**
Weight (g)	402.00 ± 30.75	492.00 ± 37.76 *	468.00 ± 29.42 *	486.00 ± 54.94 *
Liquid consumption (mL/day)	30.42 ± 4.56	44.85 ± 6.72 *	46.28 ± 6.94 *	31.00 ± 4.650 **
Liquid consumption (mL/day/100 g)	7.56 ± 0.57	9.11 ± 0.69 *	9.89 ± 0.62 *	6.37 ± 0.90 **
Equivalent in kcal in drinkable water	0.00	102.10 ± 7.83 *	110.70 ± 6.96 *	71.44 ± 10.13 * **
Feed consumption (g/day)	23.14 ± 3.47	12.14 ± 1.82 *	9.57 ± 1.43 *	9.57 ± 1.43 *
Feed consumption (g/day/100 g)	17.84 ± 1.36	7.65 ± 0.58 *	6.34 ± 0.39 * **	6.10 ± 0.86 * **
Equivalent in kcal in feed	5.75 ± 0.44	2.46 ± 0.18 *	2.04 ± 0.12 *	1.96 ± 0.27 *
Total Kcal/day/100 g body weight	17.84 ± 1.36	109.7 ± 8.42 *	117.10 ± 7.36 *	77.54 ± 11.00 * **
**Week 4**	**CG**	**OG**	**PK**	**MET**
Weight (g)	423.10 ± 30.20	527.00 ± 45.30 *	487.00 ± 42.80 #	489.00 ± 55.20 #
Liquid consumption (mL/day)	33.11 ± 4.96	48.91 ± 7.33 *	44.57 ± 6.68 *	52.97 ± 8.54 *
Liquid consumption (mL/day/100 g)	7.82 ± 0.63	9.28 ± 0.75 *	9.15 ± 0.61 *	10.83 ± 1.61 *
Equivalent in kcal in drinkable water	0.00	103.93 ± 8.40 *	102.55 ± 6.85 *	121.32 ± 18.09 *
Feed consumption (g/day)	24.42 ± 3.66	11.71 ± 1.75 *	10.71 ± 1.60 *	9.57 ± 1.43 *
Feed consumption (g/day/100 g)	5.77 ± 0.46	2.22 ± 0.17 *	2.19 ± 0.14 *	1.96 ± 0.27 *
Equivalent in kcal in feed	17.89 ± 1.44	6.88 ± 0.55 *	6.81 ± 0.45 *	6.09 ± 0.84 *
Total Kcal/day/100 g body weight	17.89 ± 1.44	110.81 ± 8.95 *	109.36 ± 10.34 *	127.41 ± 12.94 *

Values expressed as means ± SDs. (*n* = 6). * Indicates a significant difference (*p* ˂ 0.05) with respect to CG; ** indicates a significant difference (*p* ˂ 0.05) with respect to OG; # indicates that the value does not differ significantly with respect to either CG or OG. CG: control group, OG: obese rats, PK: obese rats that received extract, and MET: obese rats that received metformin.

## Data Availability

The original contributions presented in the study are included in the article. Further inquiries can be directed to the corresponding authors.

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
