# Peer review of "Extraction, Characterization, and Nutraceutical Potential of Prosthechea karwinskii Orchid for Insulin Resistance and Oxidative Stress in Wistar Rats"

_foods, 2024, doi:10.3390/foods13152432_

Round 1

Reviewer 1 Report

Comments and Suggestions for Authors

my comments are attached

Comments on the Quality of English Language

Author Response

Revisor 1

On behalf of all the authors of this manuscript, we thank you for taking the time to review it. We would also like to thank you for your suggestions and comments that help us to improve our work.

Abstract: you should first draft it like this Background (2 to 3 sentences), Scope and approach (2 to 4 sentences), Key findings. This should cover >60% of the abstract. Conclusions (2 to 3 sentences).  After you draft it like this, you remove the sections (background, scope & approach, key findings, and conclusions) and merge all the sentences.

We are grateful for your suggestions, which have been taken into account and the abstract has been redrafted in accordance with your recommendations. The abstract with the changes can be found in the corresponding section of the manuscript.

Provide a list of abbreviations.

Thank you for your suggestion, it has been taken into account and the list of abbreviations has been added to the manuscript.

List of abbreviations

SOD: Superoxide dismutase, CAT: Catalase, HOMA-IR: Homeostatic Model Assessment of Insulin Resistance, TyG: Triglyceride-glucose index, UHPLC-ESI-qTOF-MS/MS: Ultra-high performance liquid chromatography system, combined with an mass spectrometer with electrospray ionization and quadrupole time-of-flight, RI: Insulin resistance, ROS: Reactive oxygen species, NADPH oxidase: Nicotinamide adenine dinucleotide phosphate oxidase, NO: Nitric oxide, PBS: Phosphate-buffered saline, OG: Obese group, CG: Control group, PK: Group administered with P. karwinskii extract at a dose of 300 mg/kg, MET: Group administered with metformin at a dose of 200 mg/kg, ANOVA: Analysis of variance, NADH: Nicotinamide adenine dinucleotide.

Obtaining the extract: How did you maintain the ultrasonic temperature? This is very important

Thank you very much for your comments, the extraction temperature was maintained at 40 °C with a water bath. This information was added to the description of the methodology for obtaining the extract.

Compound identification with UHPLC-ESI-qTOF-MS/MS: What was the column temperature?

The column temperature for the UHPLC-ESI-qTOF-MS/MS analysis was 25 °C, this information was added to the analysis part of the manuscript.

How did you ensure that there was no false positive

We thank you for the question, it is very common for the tentative identification of compounds during the analysis with UHPLC-ESI-qTOF-MS/MS equipment due to the large number of compounds that exist and because there are not always standards to make a quick identification by only comparing the retention time during the analysis and the same fragmentation pattern when analyzed at the same conditions.

Such as the analysis performed by Ragheb et al., (2023), who performed a tentative identification of compounds by comparing their masses, fragmentation pattern, and molecular formula with those in the literature supported by the molecular network and GNPS spectral libraries (NIST14, MoNA, and Respect). Zhao et al., (2022) identified some compounds using reference standards, as well as tentative identification of other compounds based on a comparison of data obtained in previous publications, as well as with the fragmentation pattern reported in previous studies. Dahibhate et al. (2022) also performed tentative identification of some compounds from MS and MS/MS data compared with the mzCloud and ChemSpider database, as well as by comparison with the literature. Ma et al. (2023) also performed identification of some compounds by comparing retention time and MS and MS/MS information with reference standards, while tentative identification of others was performed using diagnostic fragmentation pathways, literature comparison, and comparison with GNPS and HMDB databases.

There are several points to consider when performing the tentative identification of compounds to avoid misidentification, among the things we must take into account that we must compare with studies where they analyzed standards with similar equipment to the one we use, at similar analytical conditions, with the same type of ionization, as well as the collision energy used. The compounds must have an error of less than 10 ppm and have a similar fragmentation pattern concerning the ratio of fragments and major fragments upon ionization with similar collision energies with a similar type of ionization. All this was taken into account when performing the tentative identification of the compounds present in the extract.

Parameters evaluated: What kind of anesthesia was used?

The anaesthetic used during the experimentation was sodium pentobarbital. This information was added to the corresponding section of the methodology.

Results: The science and chemistry behind your results need to be added. Also, more comparison with various literature should be added.

We are grateful for their recommendations, these were addressed and we expanded the explanation on which our results are based, mainly with respect to Table 1. We also add further comparison with the literature throughout the Results and Discussion section of the manuscript.

Table 2: Run statistical analysis and incorporate it in the in-text. The sample applies to the other tables

We appreciate your suggestion, the statistical analysis was applied to both tables.

Conclusions: Also talk about future studies

Thank you very much for your recommendation, it has been taken into account and the suggested information has been added to the conclusion section.

Therefore, future research concerning the orchid P. karwinskii will be directed towards the study of the properties of the extract as an ingredient, as well as in the design of functional foods and nutraceuticals containing its properties.

References

Dahibhate, N.L.; Dwivedi, P.; Kumar, K. GC–MS and UHPLC-HRMS Based Metabolite Profiling of Bruguiera Gymnorhiza Reveals Key Bioactive Compounds. South African Journal of Botany 2022, 149, 1044–1048, doi:10.1016/j.sajb.2022.02.004.

Ma, T.; Lin, J.; Gan, A.; Sun, Y.; Sun, Y.; Wang, M.; Wan, M.; Yan, T.; Jia, Y. Qualitative and Quantitative Analysis of the Components in Flowers of Hemerocallis Citrina Baroni by UHPLC–Q-TOF-MS/MS and UHPLC–QQQ-MS/MS and Evaluation of Their Antioxidant Activities. Journal of Food Composition and Analysis 2023, 120, 105329, doi:10.1016/j.jfca.2023.105329.

Ragheb, A.Y.; Masoud, M.A.; El Shabrawy, M.O.; Farid, M.M.; Hegazi, N.M.; Mohammed, R.S.; Marzouk, M.M.; Aboutabl, M.E. MS/MS-Based Molecular Networking for Mapping the Chemical Diversity of the Pulp and Peel Extracts from Citrus Japonica Thunb.; in Vivo Evaluation of Their Anti-Inflammatory and Anti-Ulcer Potential. Scientific African 2023, 20, e01672, doi:10.1016/j.sciaf.2023.e01672.

Zhao, M.; Linghu, K.G.; Xiao, L.; Hua, T.; Zhao, G.; Chen, Q.; Xiong, S.; Shen, L.; Yu, J.; Hou, X.; et al. Anti-Inflammatory/Anti-Oxidant Properties and the UPLC-QTOF/MS-Based Metabolomics Discrimination of Three Yellow Camellia Species. Food Research International 2022, 160, 111628, doi:10.1016/j.foodres.2022.111628.

Reviewer 2 Report

Comments and Suggestions for Authors

Dear Authors,

I revised the paper “Extraction, characterization, and nutraceutical potential of Prosthechea karwinskii on insulin resistance and oxidative stress in Wistar rats” submitted to Foods.

The paper matches the aim and scope of the Journal. It is well written. The abstract defines the background of the study, briefly describes the study and reports the main results. The introduction defines the purpose of the work and guides the reader to the logical flow of the study. However, it might be improved. The section Results and the section Discussion should be improved. Conclusions are supported by experimental data. Up-to-date references were used.

Minor revisions are needed before publication. Please, see the following comments.

Title

I suggest adding “orchid” before “Prosthechea karwinskii”.

Introduction

Lines 54-63: please, report additional key publications on this topic and highlight the novelty of the study.

Materials and methods

Paragraph 2.2: plant material was collected in 2017. When were the leaf extract prepared? How was plant material stored until extraction and analysis?

Results

Figure 1a): are PK and MET different from CG? If so, please, be consistent with Figure 1b.

Figure 1c) are PK and MET different from CG? If so, please, be consistent with Figure 1b.

Figure 1c) are PK and MET different from CG? If so, please, be consistent with Figure 1b.

Lines 163-167: are these results comparable to results from other studies available in scientific literature? Please, amend the discussion section.

Lines 179-181: how do you explain this?

Discussion

Results reported in Figure 1 and in Table 3 have not been discussed. Please, amend.

Author Response

Revisor 2

Dear Authors,

I revised the paper “Extraction, characterization, and nutraceutical potential of Prosthechea karwinskii on insulin resistance and oxidative stress in Wistar rats” submitted to Foods.

The paper matches the aim and scope of the Journal. It is well written. The abstract defines the background of the study, briefly describes the study and reports the main results. The introduction defines the purpose of the work and guides the reader to the logical flow of the study. However, it might be improved. The section Results and the section Discussion should be improved. Conclusions are supported by experimental data. Up-to-date references were used. Minor revisions are needed before publication. Please, see the following comments.

On behalf of all the authors of this manuscript, we thank you for taking the time to review it. We would also like to thank you for your suggestions and comments that help us to improve our work.

Title

I suggest adding “orchid” before “Prosthechea karwinskii”.

We thank you for your suggestion, it was taken into account and the word orchid was added to the title of the manuscript.

Introduction

Lines 54-63: please, report additional key publications on this topic and highlight the novelty of the study.

We appreciate the recommendation, it was attended and other key publications on the field were reported on, highlighting the novelty of the study.

Materials and methods

Paragraph 2.2: plant material was collected in 2017. When were the leaf extract prepared? How was plant material stored until extraction and analysis?

Thank you very much for your observation. The plant material was collected in April 2017 during the Semana Santa celebrations of that year.  The leaves were dehydrated whole and stored in paper bags at room temperature away from light and humidity until extraction. The extract was obtained in 2018, stored in an amber glass bottle that was sealed and kept at freezing temperature until analysis to avoid degradation of its compounds. This information was added to the section on plant material and extraction in the manuscript.

Results

Figure 1a): are PK and MET different from CG? If so, please, be consistent with Figure 1b.

Thank you very much for your observation and the question. In the case of Figure 1a, no significant difference was observed between the PK and MET groups compared to the control group, for that reason the asterisk was not added as superscripts in those groups. In Figure 1b, if a significant difference was observed in OG, PK, and MET compared to CG, therefore the asterisk covers all 3 groups, when compared to the OG group, only PK showed a significant difference between treatments. The description of the figures in the manuscript was worked on to clarify this part.

Figure 1c) are PK and MET different from CG? If so, please, be consistent with Figure 1b.

Thank you very much for your observation and the question, in the case of Figure 1c, there was also no significant difference between the PK and MET groups compared to the control group, which is why the asterisk was not added as superscripts in those groups. In Figure 1B, if a significant difference was observed in OG, PK, and MET compared to CG, therefore the asterisk covers all 3 groups, when compared to the OG group, only PK showed a significant difference between treatments. The description of the figures in the manuscript was reworked to make this part clearer.

Figure 1c) are PK and MET different from CG? If so, please, be consistent with Figure 1b.

Thank you very much for your observation and the question, in the case of Figure 1c, there was also no significant difference between the PK and MET groups compared to the control group, which is why the asterisk was not added as superscripts in those groups. In Figure 1B, if a significant difference was observed in OG, PK, and MET compared to CG, therefore the asterisk covers all 3 groups, when compared to the OG group, only PK showed a significant difference between treatments. The description of the figures in the manuscript was reworked to make this part clearer.

Lines 163-167: are these results comparable to results from other studies available in scientific literature? Please, amend the discussion section.

We thank you for your suggestion, it was considered and references to comparable studies in the literature were added to the discussion. This information can be found in section 3 of Results and Discussion.

Lines 179-181: how do you explain this?

We would be very grateful if you could explain why what is described in lines 179-181 happens, where it is mentioned that the activity of antioxidant enzymes increased in the obese group compared to the control group, and with the administration of the extract the activity of these enzymes decreased again to values similar to those of the control group.

According to most of the studies carried out so far, antioxidant enzyme activity decreases under oxidative stress conditions, however, in some reported cases, especially in more recent studies, antioxidant enzyme activity increases in response to oxidative stress (Leghi et al, 2015; Aouacheri et al., 2015; Hussain et al., 2017; Chen et al., 2020; Barbosa et al., 2021; Mousavi et al., 2022; Barbosa et al., 2024; Zeng et al., 2024), as occurred in the present investigation.

The model we evaluated is of rats with obesity and insulin resistance, where there is an increase in glucose levels. Hyperglycaemia causes alterations in mitochondrial morphology, increases cellular oxidation of glucose and production of nicotinamide adenine dinucleotide (NADH), which contributes to oxidative cellular processes, and in response increases the generation of reactive oxygen species (ROS) (Andreadi et al., 2022). Normally ROS are produced in cells during normal aerobic metabolism and are scavenged by antioxidant enzymes, including SOD and CAT, which protect cells against oxidative stress by cellular detoxification of O2- and H2O2, excessive ROS production and/or poor antioxidant capacity can lead to oxidative stress (Hsouna et al., 2019). In response to hyperglycemia, different metabolic signaling pathways are activated and the antioxidant system is activated to reduce ROS production and oxidative stress (Andreadi et al., 2022).

Different authors explain this increase in antioxidant enzyme activity in different ways. Barbosa et al. (2024) explain it as a compensatory mechanism for the increase in ROS due to oxidative stress. Zeng et al. (2024) consider antioxidant enzymes to be a non-specific adaptive mechanism that protects against oxidative damage and is the first line of antioxidant defense against oxidative stress. Aouacheri et al., (2015), explain that increased antioxidant enzyme activity stimulates the cellular capacity to scavenge and limit damage caused by ROS. Acting as an adaptive response and a compensatory mechanism to detoxify harmful metabolites related to oxidative stress (Aouacheri et al., 2015; Chen et al., 2020). According to Leghi et al. (2015), increased SOD activity counteracts the overproduction of O2- radicals.

Defensive mechanisms are also developed in plants to minimize or protect against the toxic effects of ROS by maintaining a balance between their synthesis and decomposition to preserve cellular redox hemostasis, one such mechanism is to modulate the activity of antioxidant enzymes (Mousavi et al., 2022).

Therefore, due to the high glucose levels in rats of the model evaluated, ROS increased in the obese group, giving an adaptive or compensatory response an increase in antioxidant enzymes to neutralize the ROS formed. When P. karwinskii extract was administered to the rats, the levels of glucose and therefore ROS decreased and consequently the levels of antioxidant enzymes did not increase.

This information can be found in the Results and Discussion section of the manuscript.

Discussion

Results reported in Figure 1 and in Table 3 have not been discussed. Please, amend.

We thank you very much for your suggestion, we have expanded the explanation of the results in Figure 1 and Table 3 and added the missing discussion. The information can be found in section 3 of Results and Discussion.

REFERENCES

Andreadi, A.; Bellia, A.; Di Daniele, N.; Meloni, M.; Lauro, R.; Della-Morte, D.; Lauro, D. The Molecular Link between Oxidative Stress, Insulin Resistance, and Type 2 Diabetes: A Target for New Therapies against Cardiovascular Diseases. Current Opinion in Pharmacology 2022, 62, 85–96.

Aouacheri, O.; Saka, S.; Krim, M.; Messaadia, A.; Maidi, I. The Investigation of the Oxidative Stress-Related Parameters in Type2 Diabetes Mellitus. Canadian Journal of Diabetes 2015, 39, 44–49, doi:10.1016/j.jcjd.2014.03.002.

Barbosa, P.O.; Souza, M.O.; Silva, M.P.S.; Santos, G.T.; Silva, M.E.; Bermano, G.; Freitas, R.N. Açaí (Euterpe Oleracea Martius) Supplementation Improves Oxidative Stress Biomarkers in Liver Tissue of Dams Fed a High-Fat Diet and Increases Antioxidant Enzymes’ Gene Expression in Offspring. Biomedicine and Pharmacotherapy 2021, 139, 111627, doi:10.1016/j.biopha.2021.111627.

Barbosa, P.O.; Tanus-santos, J.E.; Cavalli, R.D.C.; Bengtsson, T.; Barbosa, P.O.; Tanus-santos, J.E.; Cavalli, R.D.C.; Bengtsson, T. The Nitrate-Nitrite-Nitric Oxide Pathway : Potential Role in Mitigating Oxidative Stress in Hypertensive Disorders of Pregnancy The Nitrate-Nitrite-Nitric Oxide Pathway : Potential Role in Mitigating Oxidative Stress in Hypertensive Disorders of Pregnancy. 2024, doi:10.20944/preprints202404.0781.v1.

Chen, Z.; Tian, R.; She, Z.; Cai, J.; Li, H. Role of Oxidative Stress in the Pathogenesis of Nonalcoholic Fatty Liver Disease. Free Radical Biology and Medicine 2020, 152, 116–141, doi:10.1016/j.freeradbiomed.2020.02.025.

Hussain, I.; Siddique, A.; Ashraf, M.A.; Rasheed, R.; Ibrahim, M.; Iqbal, M.; Akbar, S.; Imran, M. Does Exogenous Application of Ascorbic Acid Modulate Growth, Photosynthetic Pigments and Oxidative Defense in Okra (Abelmoschus Esculentus (L.) Moench) under Lead Stress? Acta Physiologiae Plantarum 2017, 39, 1–13, doi:10.1007/s11738-017-2439-0.

Zeng, L.; Wang, Y.H.; Ai, C.X.; Zhang, B.; Zhang, H.; Liu, Z.M.; Yu, M.H.; Hu, B. Differential Effects of Oxytetracycline on Detoxification and Antioxidant Defense in the Hepatopancreas and Intestine of Chinese Mitten Crab under Cadmium Stress. Science of the Total Environment 2024, 930, 172633, doi:10.1016/j.scitotenv.2024.172633.

Reviewer 3 Report

Comments and Suggestions for Authors

The main purpose of this work was to evaluate the effect of Prosthechea karwinskii leaves extract on antioxidant enzymes superoxide dismutase (SOD) and catalase (CAT) in a model of obese rats with insulin resistance.

In my opinion this article is well written, and it presents original and relevant results about Prosthechea karwinskii leaves extract.

The Introduction contains updated references.

The methodology is correctly explained and referenced.

In item 2.7, the two equations mentioned in the text should be better described and enumerated.

Items 3 and 4 could be merged as item 3. Results and Discussions, to facilitate understanding of the research.

Author Response

Revisor 3

The main purpose of this work was to evaluate the effect of Prosthechea karwinskii leaves extract on antioxidant enzymes superoxide dismutase (SOD) and catalase (CAT) in a model of obese rats with insulin resistance. In my opinion this article is well written, and it presents original and relevant results about Prosthechea karwinskii leaves extract. The Introduction contains updated references. The methodology is correctly explained and referenced.

We thank you very much for taking the time to review our manuscript, this helps us to improve our work. The suggestions were taken into account and applied to the manuscript.

In item 2.7, the two equations mentioned in the text should be better described and enumerated.

We thank you for your suggestion, take it into account and enumerate the equations in the manuscript.

Items 3 and 4 could be merged as item 3. Results and Discussions, to facilitate understanding of the research.

We are grateful for your suggestion, it was taken into account and we merged the results section with the discussion.

Round 2

Reviewer 1 Report

Comments and Suggestions for Authors

Accept

Comments on the Quality of English Language

Minor